# Body height in young adult men and risk of dementia later in adult life

**Terese Sara Høj Jørgensen[1,2]\*, Gunhild Tidemann Okholm[1,2], Kaare Christensen[3], Thorkild IA Sørensen[2,4], Merere Osler[1,2]**

[1]Center for Clinical Research and Prevention, Bispebjerg and Frederiksberg Hospital, Copenhagen, Denmark; [2]Department of Public Health, Faculty of Health and Medical Sciences, University of Copenhagen, Copenhagen, Denmark; [3]Danish Aging Research Center, Department of Public Health, University of Southern Denmark, Copenhagen, Denmark; [4]Novo Nordisk Foundation Center for Basic Metabolic Research, Faculty of Health and Medical Sciences, University of Copenhagen, Copenhagen, Denmark

**Abstract** This study examined the relationship between body height and dementia and explored the impact of intelligence level, educational attainment, early life environment and familial factors. A total of 666,333 men, 70,608 brothers, and 7388 twin brothers born 1939–1959 and examined at the conscript board were followed in Danish nationwide registers (1969–2016). Cox regression models were applied to analyze the association between body height and dementia. Within-brothers and within-twin pair analyses were conducted to explore the role of shared familial factors including partly shared genetics. In total, 10,599 men were diagnosed with dementia. The association between one z-score difference in body height and dementia (HR: 0.90, 95% CI: 0.89;0.90) was inverse and weakened slightly after adjustment for intelligence test scores and educational level. The associations persisted in within-brother analysis and revealed a stronger, but less precise, point estimate than the cohort analysis of brothers. The twin analysis showed similar, but imprecise estimates.

\*For correspondence:
Tshj@sund.ku.dk

**Competing interests:** The authors declare that no competing interests exist.

## Introduction

Dementia poses substantial challenges for individuals and societies worldwide. (*Livingston et al., 2017*) This has motivated studies of potential predictors and risk factors for dementia in recent years. (*Livingston et al., 2017*) Development of dementia may be a result of both genetics and environmental exposures operating throughout the life course. (*Borenstein et al., 2006*; *Bird, 2005*) A newly published meta-analysis and systematic review showed that the risk of dementia may already be established early in life. (*Wang et al., 2019*) Body height has a strong genetic component and is at the same time influenced by environmental factors such as childhood diseases and nutrition. (*Jelenkovic et al., 2016a*; *Jelenkovic et al., 2016b*) Short height has been linked to development of dementia in a number of smaller studies (N = 203–3,734) and to dementia as cause of death in a large study pooling 18 prospective cohorts (N = 181,800). (*Russ et al., 2014*; *Petot et al., 2007*; *Abbott et al., 1998*; *Gatz et al., 2006*; *Huang et al., 2008*; *Beeri et al., 2005*) Body growth may be related to dementia as an indicator of brain and cognitive reserve and corresponding individual differences in the brain structure, which could imply differences in individual resilience toward development of dementia. (*Wang et al., 2019*) Another possible explanation of the height-dementia association is the correlation between body height and level of growth hormone that through hippocampal function and cognition has been linked to the risk of developing dementia. (*Borenstein et al., 2006*; *Abbott et al., 1998*; *Beeri et al., 2005*) Thus, rather than being a risk factor in itself, short body height is likely an indicator of harmful exposures early in life. (*Russ et al.,*

*2014*) However, large scale high-quality longitudinal studies exploring the impact of early environmental factors and genetics to explain the link between body height and dementia are needed. (*Borenstein et al., 2006*; *Wang et al., 2019*; *Russ et al., 2014*) Previous studies question whether body height and dementia are solely linked by brain and cognitive reserve measured by intelligence and socioeconomic factors, or by shared underlying familial factors including genetics that may influence both body height and dementia. In this study, we examined whether body height in young adult men is associated with diagnosis of dementia while exploring whether intelligence test score, educational level, and underlying environmental and genetic factors shared by brothers explain the relationship.

## Results

*Figure 1* presents the selection of the three study populations; 1) men (N = 666,333), 2) brothers (N = 70,608), and 3) twins (N = 7,388). The total study population of 666,333 men were followed for an average of 41.4 years from a baseline mean age of 22.1 years (due to delayed entry for some men) to a maximum age of 57–77 years. The mean age at measurement of height was 19.73 years (95% confidence interval (CI):19.73;19.74). *Supplementary file 1* provides body height measures and information on dementia diagnoses of the total population of men. In total, 10,599 (1.6%) men were registered with dementia diagnosis during follow-up. The mean height was 176.8 cm (SD: 6.6). The Table furthermore shows that the mean height was greater among men with higher intelligence level, higher level of education, younger birth cohorts, and those living in the Capital region. Similarly, these groups also experiences fewer cases of dementia.

In Model 1, the estimated hazard ratio (HR) was 0.86 (95%CI:0.85–0.87), which attenuated slightly after the additional adjustments for educational level and intelligence test scores in Model 2a and 2b (*Table 1*). The fully adjusted estimate of the birth cohort-specific z-score of body height (Model 3) showed that for each one unit taller z-score of body height, the HR of dementia were 0.90 (95% CI:0.89;0.90). This was the same estimate as found for the model adjusted for intelligence test scores

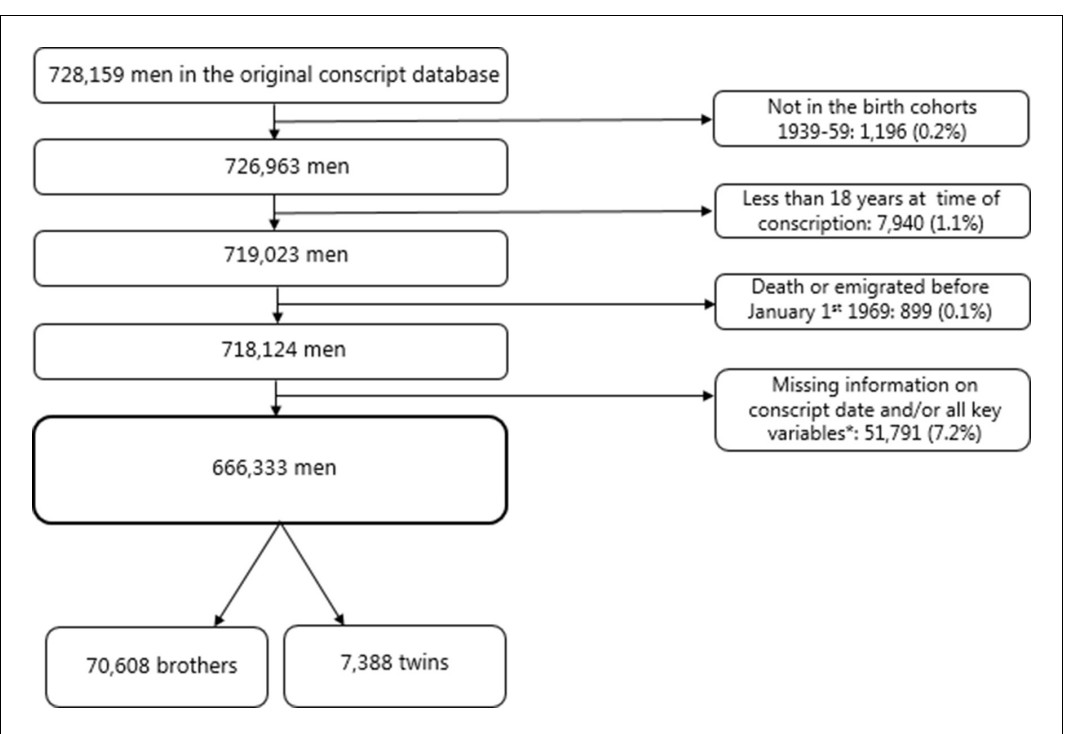

**Figure 1.** Selection of the study population.
The online version of this article includes the following figure supplement(s) for figure 1:

**Figure supplement 1.** Time line of data collection from each of the registers and the total follow-up time.

**Table 1.** Hazard ratios (HRs) and corresponding 95% confidence intervals (95% CIs) of the association between taller body height at entry to adulthood and dementia diagnoses among the total population of men.

| Descriptive statistics | | HR (95% CI) for onset of dementia per one z-score* higher | | | |
|---|---|---|---|---|---|
| Mean height (Standard diviation) | Dementia cases Number (%) | Model 1[†] | Model 2a[‡] | Model 2b[§] | Model 3[¶] |
| 176.8 (6.6) | 10,599 (1.6) | 0.86 (0.85;0.87) | 0.88 (0.87;0.89) | 0.90 (0.89;0.90) | 0.90 (0.89;0.90) |

*Identify cohort-specific values of one z-score in **Supplementary file 2**.

[†]Model 1: Stratified by birthcohort and conscript board district. Age included as underlying scale of the model.

[‡]Model 2a: Model 1 + adjusted for educational level.

[§]Model 2b: Model 1 + adjusted for intelligence test scores.

[¶]Model 3: Model 1 + adjusted for educational level and intelligence test scores.

without educational level (Model 2b). **Supplementary file 2** shows that the mean heights range from 174.5 to 179.1 cm and the z-scores range from 6.34 to 6.6 cm for the birth cohorts in the total population without any clear patterns of change over time. The finding of 10% lower HR with one z-score higher body height suggests that men born in 1939 with a body height of 181.5 cm (equal to one z-score taller than the mean) can expect a 10% lower dementia risk compared with men with the mean body height (175.0 cm). The corresponding body height for the latest birth cohort (1959) is 185.6 cm compared with the mean body height of 179.1 cm. **Figure 2** shows that the association between z-scores of body height with dementia diagnosis is curve-linear; the HRs showed a slightly stronger increase with lower z-scores of body height below the z-score reference of 0 than above the z-score reference of 0.

Analyses of Model 3 with the risk period divided at age less than 60 years and 60 years and older showed that the association between one z-score taller body height and dementia was slightly smaller among men below 60 years (HR:0.87, 95% CI: 0.84;0.90) than men 60 years and older (HR:0.91, 95% CI: 0.90;0.92) (**Table 2**).

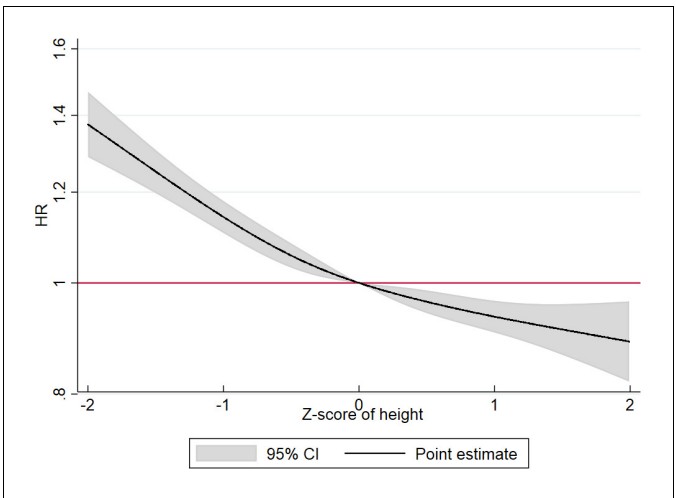

**Figure 2.** Hazard Ratios (HRs) and corresponding 95% confidence intervals (95% CIs) for the association between z-score of body height at the entry to adulthood as a cubic spline with four knots and dementia. A z-score of 0 is the reference value. Stratified by birth cohort and adjusted for conscript board district, educational level and intelligence test scores. Age included as underlying scale of the model. The analysis included the total population of men. Identify cohort-specific values of one z-score in **Supplementary file 2**.

The online version of this article includes the following figure supplement(s) for figure 2:

**Figure supplement 1.** Distribution of mean numeric difference in body height in cm between brothers (**A**) and twins (**B**).

**Table 2.** Fully adjusted* hazard ratios (HRs) and corresponding 95% confidence intervals (95% CIs) of the association between taller body height at entry to adulthood and dementia diagnosis among all men.

| Dementia cases Number (%) | HR (95% CI) for onset of dementia per one z-score* higher |
|---|---|
| <60 years | |
| 4191 (0.6) | 0.87 (0.84;0.90) |
| ≥60 years | |
| 6408 (1.3) | 0.91 (0.90;0.92) |

*Identify cohort-specific values of one z-score in **Supplementary file 2**.

†Model 3: stratified by birth cohort and adjusted for conscript board district, educational level and intelligence test scores. Age included as underlying scale of the model.

Analyses of Model 3 with the follow-up period divided by 1) prior to 1995 and 2) 1995 and thereafter showed that the association between one z-score taller body height and dementia was comparable between the two periods (**Supplementary file 3**). The numeric mean (SE) body height difference was 5.1 (0.02) cm between brothers and 3.5 (0.06) cm between twins (See **Figure 2—figure supplement 1** for distribution of the numeric difference in body height for brothers (A) and twins (B)). The analyses of Model 3 showed that the within-pair analysis of brothers revealed a stronger, but less precise, point estimate than the cohort analysis of brothers (**Table 3**). For twins, the comparison of the cohort analysis of twins and the within-pair analysis of twins revealed a pattern similar to that found for brothers. Yet, the pattern was less clear and the HRs were imprecisely estimated possible due to a lower number of individuals in these analyses (N = 7,388) (**Table 3**).

## Discussion

In this nationwide cohort of Danish men born from 1939 through 1959, we detected that taller body height at the entry to adulthood was associated with lower risk of dementia diagnosis later in life. The relationship attenuated, but persisted after adjustments for educational level and intelligence test scores. Within-brother analysis confirmed the findings of a relationship between taller body height and lower risk of dementia diagnosis, and even revealed a stronger, but less precise, point estimate of the association than the cohort analysis of brothers. The findings from the within-twin analyses were in the same direction, but with more uncertainty. These findings support that the relationship between body height and dementia is not explained by shared genetics and exposures to other familial factors shared between brothers.

**Table 3.** Fully adjusted* hazard ratios (HRs) and corresponding 95% confidence intervals (95% CIs) of the association between taller body height at entry to adulthood and dementia diagnosis among brothers and twins

| Descriptive statistics | | HR (95% CI) for onset of dementia per one z-score* higher | |
|---|---|---|---|
| Mean height (standard deviation) | Dementia cases Number (%) | Cohort analyses | Within-brother/twin analyses |
| Brothers | | | |
| 177.8 (6.6) | 597 (0.8) | 0.90 (0.82;0.98) | 0.78 (0.64;0.96) |
| Twins | | | |
| 175.8 (6.7) | 107 (1.4) | 0.91 (0.73;1.16) | 0.86 (0.44;1.68) |

*Identify cohort-specific values of one z-score in **Supplementary file 2**.

†Model 3: Stratified by birth cohort and adjusted for conscript board district, educational level and intelligence test scores. Age included as underlying scale of the model.

## Strengths and limitations

This study has a number of strengths. It is based on a very large unselected population followed from early adulthood until a maximum age of 57–77 years for dementia diagnoses. This study design is very important as previous findings may have underestimated the relationship due to selection of the healthiest part of the population, who are willing to participate, and have survived into this age. It is also a great advantage that we were able to investigate associations between body height and dementia adjusted for intelligence test scores and educational level. Intelligence together with educational level have been shown to be strong markers of cognitive reserve, which are hypothesized to be highly predictive of the age at which dementia is manifested and diagnosed. (*Borenstein et al., 2006*) We further included objective measures of body height rather than self-reported height with its implicit reporting errors. In the same line, this study included measures of body height from early adulthood, whereas previous studies have used measures from mid- and late-life. This could imply a bias because declining height since young adulthood (*Cline et al., 1989*) has been found to be more pronounced in socioeconomic dis-advantaged groups and those with lower later-life cognitive function. (*Huang et al., 2013*) Further, underdiagnoses of dementia has in some countries been shown to be greater in socioeconomically disadvantaged groups. (*Lang et al., 2017*) A final strength is that we were able to investigate the impact of familial factors, including various degrees of shared genetics and environmental factors, by conducting within-brother and twin-pair analyses.

We acknowledge the limitations of the study. A potential concern is that even though dementia diagnoses in the Danish National Patient Register is likely to be correct, (*Phung et al., 2007*; *Salem et al., 2013*) under-diagnosis is possible. The registers and identifications to define dementia change over time (*Figure 1—figure supplement 1*) and this difference in coverage could lead to differences in diagnostic validity and completeness at different time points during follow-up. However, we believe this is a minor issue as a previous study of data from the the Danish Conscription Database (DCD), which investigated the relationship between cognitive ability and dementia, only found small differences in risk estimates when using international classification of disease (ICD) version eight and ICD10 codes, respectively, to identify dementia. (*Osler et al., 2017*) Furthermore, the introduction of the Danish Prescription Registry on January 1st 1995 has enabled identification of dementia patients who are not in contact with the secondary healthcare system for their dementia diagnosis, but who receive medication through their general practitioner. This has most likely improved the sensitivity of the dementia diagnoses from 1995. Our sensitivity analyses showed comparable findings of the relationship between body height and dementia before and after the introduction, yet the comparison of these results were limited by including men in different age-groups (<1995: maximum age of 36–56 and ≥1995: maximum age of 57–77). Another issue is that the recorded diagnosis of dementia has been shown to be less valid in younger populations (*Phung et al., 2007*; *Salem et al., 2013*) and our model tests indicated that the HRs were slightly less pronounced at higher ages. We therefore conducted analyses where follow-up time was split at age less than 60 years and 60 years and older. The analyses of both age-groups showed associations between taller body height and lower risk of dementia, but the point estimate was slightly stronger in the younger population. This may partly be explained by the less valid symptom profile in these patients and misdiagnoses of psychiatric or neurological conditions for dementia. (*Salem et al., 2013*) Also, the findings may not be generalizable to men aged above 77 years as the study population only reached a maximum age of 57–77 years during the follow-up period. Furthermore, the role of familial factors were not fully explored. The analyses of brothers were incomplete by 73.5% of the population being from the birth cohorts 1953–59 and by being restricted to brothers with known shared mothers, implying a mixture of twin brothers, full non-twin brothers and maternal half-brothers, but without paternal half-brothers. The specific contribution of genetic factors, independent of environmental factors, cannot be assessed by twin-pair analyses without distinction between mono- and dizygotic twins. Finally, the generalizability of the findings to women is questionable. Previous findings on potential gender differences in the relationship between body height and dementia or poor cognition in old age are inconclusive, (*Petot et al., 2007*; *West et al., 2015*) but the incidence and prevalence of dementia are found to be lower in men than in women. (*Matthews et al., 2016*).

## Findings in light of previous studies

The findings of this current study provide substantial support to previous evidence of a link between body height and dementia. (*Borenstein et al., 2006*; *Wang et al., 2019*; *Russ et al., 2014*; *Petot et al., 2007*; *Abbott et al., 1998*; *Gatz et al., 2006*; *Huang et al., 2008*; *Beeri et al., 2005*) All previous studies had accounted for educational level and other socioeconomic indicators, yet none of these studies had adjusted for intelligence level earlier in life. Intelligence level has been suggested to be a stronger marker of brain and cognitive reserve than educational level. (*Borenstein et al., 2006*) Intelligence level is furthermore correlated with body height (*Christensen et al., 2015*; *Beauchamp et al., 2011*) and by itself associated with dementia. (*Osler et al., 2017*).

In contrast to previous studies, we also investigated the impact of other potential early-life familial factors including genetics and socioeconomic resources in the family that may influence both body height and later risk of dementia. (*Wang et al., 2019*) Body height has been shown to have a strong genetic component with around 80% of the variation in populations being explained by genetic differences between individuals. (*Jelenkovic et al., 2016a*; *Lango Allen et al., 2010*; *McEvoy and Visscher, 2009*) The genetic component of height has furthermore been found to be consistent across countries independent of living standards. (*Jelenkovic et al., 2016a*).

Interestingly, we found that the estimate for the relationship between body height and dementia was stronger in the within-pair analysis than in the cohort analysis of brothers. The genetic and environmental variation influencing body height, but not risk of dementia, is smaller within brothers than between men in general, which may weaken the association between body height and dementia in the latter compared to the former group. Through this mechanism, the finding of a stronger association within brothers may be explained by less dilution of the effects of different harmful exposures early in life influencing both body growth and risk of dementia. These findings furthermore suggests that genetics has a minor role in the association of body height and dementia. However, it is important to acknowledge that the within-brother analysis was subject to greater uncertainty shown by a wider confidence interval. One previous Swedish study applied a twin design including 106 monozygotic twin pairs discordant for dementia, which indicated that the twin with the shortest height also more often was the one who developed dementia, supporting our interpretation, but the association provided insignificant estimates. (*Gatz et al., 2006*) Two of the previous studies, which had adjusted for APOE genotype, being the strongest single genetic marker related to development of dementia, also supported that conclusion. In more detail, one of the studies detected an association between body height and dementia, (*Petot et al., 2007*) and the other study, examining length of extremities and dementia, found an association to arm length in both women and men, but only an association to knee height (i.e. distance from foot sole to the anterior surface of the thigh of the lower leg) in women. (*Huang et al., 2008*).

In conclusion, taller body height at the entry to adulthood, supposed to be a marker of early-life environment, is associated with lower risk of dementia diagnosis later in life. The association persisted when adjusted for educational level and intelligence test scores in young adulthood, suggesting that height is not just acting as an indicator of cognitive reserve. Within-brother analysis confirms the findings of a relationship between taller height and lower risk of dementia diagnosis, and suggest that the association may have common roots in early life environmental exposures that are not related to family factors shared among brothers. This is supported in within-twin analysis, although the finding was imprecisely estimated.

## Materials and methods

The study was based on the DCD, which, in brief, holds information registered at mandatory Danish conscription examinations in the years 1957 to 1984 for men born from 1939 through 1959 (*Christensen et al., 2015*). All Danish men are requested by law to appear before the conscript board for a physical and mental examination between the ages of 18 and 26 years. We linked the data with the Danish Twin Register, the Danish Civil Registration System, the Danish National Patient Registry, the Danish Psychiatric Central Register, and the Danish National Prescription Register. (*Mors et al., 2011*; *Skytthe et al., 2011*; *Pedersen, 2011*; *Kildemoes et al., 2011*; *Lynge et al., 2011*) *Figure 1* presents the selection of the three study populations; 1) men (N = 666,333), 2) brothers (N = 70,608), and 3) twins (N = 7,388). Brothers and twins were identified by two different

approaches. Brothers were identified from the population through maternal linkage in the Danish Civil Registration System and 73.5% were from the birth-cohorts 1953–1959, where the linkage between mothers and offspring is complete in the register. The coverage vary between birth cohorts because the linkage between parents and offspring was based on shared residency in 1968 where the Danish Civil Registration System was established. After 1968, the linkage was carried out at the time of birth. Please see *Supplementary file 4* for number of brother included from each birth cohort (1939–1959). Brothers were defined broadly by including all men with the same identified mother that is full-brothers, half-brothers, adopted brothers, and twins, triplets etc. Thus, the sample of brothers also include twins from these birth cohorts as they are registered with the same mother. However, for the twin population, twin pairs in all the birth cohorts 1939–1959 were identified by linking the DCD to the Danish Twin Register. The project was evaluated and approved by the Danish Data Protection Agency: Jr nr 2014-41-2911. According to Danish law, ethical approval is not required for purely register-based studies.

## Main variables

Body height at entry to adulthood was included as the main exposure in this study. Objective measures of body height without shoes at the time of conscript board examination were investigated as birth cohort specific z-scores (See *Supplementary file 2*). The z-scores were investigated as a continuous variable.

Dementia diagnosis was measured by in- and out-patient contacts with ICD8 and ICD10 codes (ICD8: 290.00–290.99 and ICD10: F00.0-F03.9; G30.0-G30.9) in the Danish Psychiatric Central Register, since 1969, and in the Danish National Patient Registry since 1977. (*Mors et al., 2011*; *Lynge et al., 2011*) Dementia diagnosis was furthermore identified by redeemed prescriptions of acetylcholinesterase inhibitor registered by Anatomic Therapeutical Chemical (ATC) codes (N06D) in the Danish National Prescription Registry since 1995. (*Kildemoes et al., 2011*) (See *Figure 1—figure supplement 1* for a graphical illustration of the data collection time line for each of these registers). The dementia diagnoses in the Danish hospital registers have been shown to have a positive predictive value of 70% and 83% in two different randomly selected populations of 200 individual with dementia recorded in the register. The values vary between the two studies because of age differences in the study populations, the lowest positive predictive value was detected for the population with the youngest age group. (*Phung et al., 2007*; *Salem et al., 2013*).

## Covariates

Intelligence test scores, educational level, age and district at conscript board examination, and birth cohorts were included as covariates in the analyses. Intelligence test scores were measured by the Børge Priens Prøve (BPP), which has been shown to be highly correlated with the full-scale Wechsler Adult Intelligence Scale score (R = 0.82) (*Reinish and Teasdale, 1989*) The score ranged from 0 to 78 and was categorized in deciles by 1)$\leq$21, 2) 22–27, 3) 28–31, 4) 32–35, 5) 36–38; 6) 39–42, 7) 43–45, 8) 46–49, 9) 50–53, 10)$\geq$54, and missing. Educational level at the time of examination was categorized by 1) short educational level: primary school, 2) medium educational level: vocational education and training as reference, 3) long educational level: secondary school, medium length education, high school, and academic educations, and 4) missing. Age was included as a continuous variable. Information about the eight conscript board examination districts was included as a categorical covariate with the following categories; 1) Copenhagen greater area incl. northern part of Zealand, 2) the remaining parts of Zealand and adjacent islands, 3) Funen and adjacent islands, 4) Mid and South parts of Jutland, 5) North and West Jutland, 6) Bornholm, 7) Southernmost part of Jutland, and 8) missing. The birth cohorts were included as a continuous variable ranging from 1939 through 1959.

## Statistical analyses

The association between body height at entry to adulthood and diagnosis of dementia was analyzed in three populations: 1) all men in the birth cohorts 1939–1959, 2) brothers mainly from the cohorts 1953–1959, and 3) twins in the birth cohorts 1939–1959.

We applied Cox proportional hazard regression models to estimate the associations between body height and subsequent dementia diagnosis (HR with 95% CI). For men with a conscript board

examination date before January 1st 1969, age at this date was used as baseline. In contrast, men examined after January 1st 1969 had baseline at the age of their conscript board examination. The mean age was 22.1 years at baseline due to the delayed entry for some of the men. Men were included in the statistical analyses at baseline and followed until diagnosis of dementia, emigration, death, or end of follow-up (April 30th 2016) that is to a maximum age of 57–77 years. Age was the underlying time scale of the model. Model assumptions of time-constant HRs were tested by Schoenfeld residuals for continuous variables and log-minus-log curves for categorical variables. The model assumption was fulfilled for all variables. However, there was a tendency that the association between body height and dementia was stronger in the younger age groups, thus, a supplementary analyses with follow-up split at 1) age <60 years and 2) age ≥60 years were conducted to quantify this potential difference in the association between body height and dementia diagnosis before and after age 60 years. Sensitivity analysis of the association 1) before 1995 (dementia diagnosis identified by ICD8 codes) and 2) 1995 and thereafter (dementia diagnosis identified by ATC codes and ICD10 codes) was performed.

To investigate the impact of adjustments for intelligence test scores and educational level as important explanatory factors, the analyses of the total population of men were conducted in four steps by Model 1 adjusted for conscript board examination district, Model 2a-b further adjusted for either educational level or intelligence test scores and finally, Model 3 further adjusted for both educational level and intelligence test scores. All the analyses were stratified by birth cohort to account for potential birth cohort effects and included a cluster term to account for the interdependence of observations between brothers. Finally, all models were by default adjusted for age as the underlying scale of the statistical models. Body height was also explored as restricted cubic splines with four knots equally distributed over the range of the z-score variable. The knots were identified at z-scores of −1.63,–0.40, 0.36 and 1.65 with 0 as the reference.

To explore the role of shared familial factors including environmental factors and partly shared genetics for the association between body height and dementia diagnosis, sibling populations (brothers and twins) were analyzed in a Model 3 standard cohort analyses and within-brother/twin pair analyses. In the cohort analyses each brother and twin was treated as an individual and a cluster term was included to account for the interdependence of observations. The within-pair analyses were stratified by a variable identifying the brother/twin pair cluster to make comparisons within groups of brothers and twin pairs. These analyses will provide estimates that, in addition to the included co-variates, are adjusted for shared family factors.

Statistical analyses were conducted in the statistical software packages Stata version 15 (Research Resource Identifier: SCR_012763).

## Acknowledgements

Authors thank D Molbo, and EL Mortensen, who together with K Christensen, M Osler and TIA Sørensen established the database. The work was supported by the Danish Medical Research Council [grant number 09–063599 and 09–069151], the Velux Foundation [grant number 31205], the Jascha Foundation, the Health Foundation (17-B-0033), Doctor Sofus Carl Emil Friis and Olga Doris Friis grant, and the Social Inequalities in Ageing (SIA) project, funded by NordForsk, project no. 74637.

## Additional information

### Funding

| Funder | Grant reference number | Author |
| --- | --- | --- |
| Danish Medical Research Council | 09-063599 | Merere Osler |
| Velux Foundation | 31205 | Kaare Christensen |
| Jascha Fonden | | Gunhild Tidemann Okholm |
| Læge Sophus Carl Emil Friis og hustru Olga Doris Friis' Legat | | Merere Osler |

| NordForsk | The Social Inequality in Ageing project (74637) | Terese Sara Høj Jørgensen |
|---|---|---|
| Danish Medical Research Council | 09-069151 | Merere Osler |
| Health Foundation | 17-B-0033 | Merere Osler |

The funders had no role in study design, data collection and interpretation, or the decision to submit the work for publication.

## Author contributions

Terese Sara Høj Jørgensen, Conceptualization, Formal analysis, Investigation, Methodology, Project administration; Gunhild Tidemann Okholm, Conceptualization, Supervision, Methodology; Kaare Christensen, Supervision, Methodology; Thorkild IA Sørensen, Conceptualization, Supervision, Investigation, Methodology; Merere Osler, Conceptualization, Data curation, Supervision, Funding acquisition, Investigation, Methodology

## Author ORCIDs

Terese Sara Høj Jørgensen  https://orcid.org/0000-0003-1450-4472
Thorkild IA Sørensen  http://orcid.org/0000-0003-4821-430X
Merere Osler  http://orcid.org/0000-0002-6921-220X

## Decision letter and Author response

Decision letter https://doi.org/10.7554/eLife.51168.sa1
Author response https://doi.org/10.7554/eLife.51168.sa2

## Additional files

### Supplementary files

• Source code 1. Stata code used for the analyses.

• Supplementary file 1. Distribution of covariates for body height and dementia diagnoses among the total population of men.

• Supplementary file 2. Mean body height (z-scores) in cm for each of the birth cohorts in the full study population, brothers and twins.

• Supplementary file 3. Fully adjusted* hazard ratios (HRs) of the association between taller body height at entry to adulthood and dementia diagnosis among all men before 1995 and from 1995 and onwards.

• Supplementary file 4. Frequency (%) of individuals included from each birth cohort in the brother analyses.

• Transparent reporting form

### Data availability

We have uploaded a file with the code used to program the statistical analyses performed in Stata. To access the anonymised dataset used in this study, researchers need to apply the Danish Conscript board steering committee and Statistics Denmark. Only aggregated data, where no identification of persons is possible i.e. minimum five observations in each cell, can be removed from the server containing the data accessed through Statistics Denmark. Thus, we cannot provide an anonymised copy of the dataset as individuals may be identified based on the information in the data e.g. birthday, height, dementia status etc. Access to the data through Statistics Denmark is only granted for authorised research and analysis environments of a more permanent nature with a chief researcher and several researchers/analysts. Foreign researchers affiliated to a Danish authorised environment can also get access. Authorisation is granted by the Director General. Please find more information in the document 'Access to micro data at Statistics Denmark_2014' on https://www.dst.dk/en/Til-Salg/Forskningsservice.

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
