## [Decision Letter]

**Acceptance summary:**

Body height, a biomarker that reflects the interplay of genetic endowment and early-life experiences and exposures (e.g., nutrition, social and psychological circumstances), may provide insights into the development of a wide range of disease outcomes. In the current study, the authors explore the association between body height and dementia using information from the Danish national registers. Their work showed that people who were taller had significantly lower rates of dementia after taking into account the effect of several other factors. Further, to distinguish this protective effect of body height from genetic predisposition, which is a strong determinant of height, the authors conducted an analysis of siblings and twins and their findings remain the same.

This study represents a promising start to new research that could shed light on the potential influence of early life risk factors for dementia since, by the very nature of the growth process, height is determined in early life.

**Decision letter after peer review:**

Thank you for submitting your article "Body height in young adult men and risk of dementia later in adult life" for consideration by *eLife*. Your article has been reviewed by three peer reviewers, and the evaluation has been overseen by Belinda Nicolau as Reviewing Editor and Eduardo Franco as Senior Editor. The reviewers have opted to remain anonymous.

They discussed the reviews with one another and submitted them to the Reviewing Editor, who has drafted this decision to help you prepare a revised submission.

Summary:

Body height, a biomarker that reflects the interplay of genetic endowment and early-life experiences and exposures (e.g., nutrition, social and psychological circumstances), may provide insights into the aetiology of a wide range of disease outcomes. In the current study, the authors explore the association between body height and dementia. They use Danish national registers to obtain data from the conscript board, including height, intelligence, and educational attainment, matched with data from health registers to identify those with dementia diagnoses or who were taking medication typically prescribed for Alzheimer's disease. Their results showed that taller body height was associated with significantly lower rate of dementia after adjusting for a series of confounders. In addition, they attempted to distinguish this protective effect of body height from confounding by genetics – a strong determinant of height – through analysis of within siblings and within twin differences. This study represents a potentially valuable contribution to the literature on early life risk factors for dementia. However, there are a few substantive concerns.

Essential revisions:

The authors underplay the importance of the analyses of siblings and twins. If height is a risk factor in intrapair analyses of siblings and twins, and comparable in magnitude to a standard cohort analysis, then it is strong evidence that the explanation for the association between height and dementia is something other than shared genes or shared rearing environment. Indeed, one of the main concerns raised by the reviewers is the adjustment for genetics with the within-twin pair analyses. The point estimate for within-twin pairs is similar to the point estimate for unrelated individuals. In the larger sample of unrelated individuals (Table 1), the confidence intervals (CI) are below 1.0, whereas the CI include 1.0 in both the cohort analysis and the twin analysis in Table 3. This likely is a reflection of the smaller N. This may, moreover, suggest that adjusting for genetics does not explain the association between body height and dementia. The authors don't reach this conclusion. In the Discussion, they state "These findings support a relationship between body height and dementia independent of the genetic relationship and of exposures to other shared early-life familial factors." The authors should elaborate on this interpretation, revisiting their reasoning so as to clarify the argument.

A second issue of concern raised is the definition of what constitutes tall height. The reviewers were concerned with the meaningful clinical interpretation of the results. They should be wary of the ability of a large study such as this to detect differences, which may not be meaningful. Even without considering the role of genetics, and taking at face value the results of the 'fully adjusted' model (Model 3) that shows a 10% reduction in dementia risk for every one z-score increase in height, how clinically meaningful is this reduction? Since your z-score is defined as how far you are from the average in a group, in the real-world, how tall should a person be to start getting any 'meaningful' protective effect from what is being reported? Clinical pertinence should be considered.

The authors discuss the limitations of obtaining dementia diagnoses from patient registers. However, they do not address the introduction of the national prescription register in 1995 and how that might affect sensitivity, specificity, and accuracy in identifying dementia cases.

The authors include birth cohort in the analyses and compare results dividing at age 60. They do not, however, address the fact that participants were aged 57 to 77 at the latest date for register dementia diagnoses, so that a large proportion are younger than average age of onset for dementia. Prevalence of dementia is quite low, which likely reflects both cases missed by registers and the age structure of the study population.

Not until the detailed methods is it learned that brothers were from 1953-1959 birth cohorts, making them age 57 – 63 as of the latest date of matching to dementia diagnoses. (The text states that the brothers were "mainly" from 1953-1959, so it's a bit ambiguous.) If the brothers are from 1953-1959 birth cohorts, it is not surprising that rate of dementia cases is lower for brothers than for the full population or for the twins. It seems possible that the difference in age structure might also contribute to the observation that the point estimate for the association between body height and dementia is greater in the twin pair analyses than in the analyses of brothers. It would also make the observation less interesting. This difference in birth cohorts is essential to note because, otherwise, the assumption would be that the sample of brothers contains all twin pairs in the sample of twins, since twins are also siblings.

It would be of interest to describe the mean and distribution of intrapair difference in height in centimeters (cm).

---

## [Author Response]

Essential revisions:The authors underplay the importance of the analyses of siblings and twins. If height is a risk factor in intrapair analyses of siblings and twins, and comparable in magnitude to a standard cohort analysis, then it is strong evidence that the explanation for the association between height and dementia is something other than shared genes or shared rearing environment. Indeed, one of the main concerns raised by the reviewers is the adjustment for genetics with the within-twin pair analyses. The point estimate for within-twin pairs is similar to the point estimate for unrelated individuals. In the larger sample of unrelated individuals (Table 1), the confidence intervals (CI) are below 1.0, whereas the CI include 1.0 in both the cohort analysis and the twin analysis in Table 3. This likely is a reflection of the smaller N. This may, moreover, suggest that adjusting for genetics does not explain the association between body height and dementia. The authors don't reach this conclusion. In the Discussion, they state "These findings support a relationship between body height and dementia independent of the genetic relationship and of exposures to other shared early-life familial factors." The authors should elaborate on this interpretation, revisiting their reasoning so as to clarify the argument.

We completely agree that the results from the brother and twin analyses are of great importance in this study. We furthermore agree that the brother analyses indicate that genetics and shared environment between brothers do not explain the relationship between body height and dementia and that the findings from the twin analyses may be subject to power issues causing imprecise estimates with wider confidence intervals including 1.0 (reference value).

In the following, we have included the paragraphs from the manuscript where we elaborate on the results from the brother and twin analyses. We have rewritten some of these paragraphs from the previous version of the manuscript to emphasize the importance of these findings:

In the Abstract: “The associations persisted in within-brother analysis and revealed a stronger, but less precise, point estimate than the cohort analysis of brothers. The twin analysis showed similar, but imprecise estimates.”

In the Results: “The analyses of Model 3 showed that the within-pair analysis of brothers revealed a stronger, but less precise, point estimate than the cohort analysis of brothers (Table 3). For twins, the comparison of the cohort analysis of twins and the within-pair analysis of twins revealed a pattern similar to that found for brothers. Yet, the pattern was less clear and the HRs were imprecisely estimated possibly due to a lower number of individuals in these analyses (N=7,388) (Table 3).”

In the Discussion:

· “Within-brother analysis confirmed the findings of a relationship between taller body height and lower risk of dementia diagnosis, and even revealed a stronger, but less precise, point estimate of the association than the cohort analyses of brothers. The findings from the within-twin analyses were in the same direction, but with more uncertainty. These findings support that the relationship between body height and dementia is not explained by shared genetics and exposures to other familial factors shared between brothers.”

· “A final strength is that we were able to investigate the impact of familial factors, including various degrees of shared genetic and environmental factors, by conducting within-brother and twin-pair analyses.”

· “Interestingly, we found that the estimate for the relationship between body height and dementia was stronger in the within-pair analysis than in the cohort analysis of brothers. The genetic and environmental variation influencing body height, but not risk of dementia, is smaller within brothers than between men in general, which may weaken the association between body height and dementia in the latter compared to the former group. Through this mechanism, the finding of a stronger association within brothers may be explained by less dilution of the effects of different harmful exposures early in life influencing both body growth and risk of dementia. These findings furthermore suggests that genetics has a minor role in the association of body height and dementia.”

· “Within-brother analysis confirms the findings of a relationship between taller height and lower risk of dementia diagnosis, and suggest that the association may have common roots in early life environmental exposures that are not related to family factors shared among brothers. This is supported in within-twin analysis, although the finding was imprecisely estimated.”

However, we believe it is important to acknowledge the limitations of the brother and twin analyses. We have therefor kept the following paragraphs in the manuscript:

In the Discussion:

· “Furthermore, the role of familial factors were not fully explored. The analyses of brothers were incomplete by 73.5% of the population being from the birth cohorts 1953-59 and by being restricted to men with known shared mothers, implying a mixture of twin brothers, full non-twin brothers and maternal half-brothers, but without paternal half-brothers. The specific contribution of genetic factors, independent of environmental factors, cannot be assessed by twin-pair analyses without distinction between mono- and dizygotic twins.”

· “…However, it is important to acknowledge that the within-brother analysis was subject to greater uncertainty shown by a wider confidence interval”

A second issue of concern raised is the definition of what constitutes tall height. The reviewers were concerned with the meaningful clinical interpretation of the results. They should be wary of the ability of a large study such as this to detect differences, which may not be meaningful. Even without considering the role of genetics, and taking at face value the results of the 'fully adjusted' model (Model 3) that shows a 10% reduction in dementia risk for every one z-score increase in height, how clinically meaningful is this reduction? Since your z-score is defined as how far you are from the average in a group, in the real-world, how tall should a person be to start getting any 'meaningful' protective effect from what is being reported? Clinical pertinence should be considered.

Thank you for this comment. We also believe it is important to highlight the clinical relevance of the findings. To be subject to a 10% lower dementia risk, a man born in 1939 should be 6.5 cm (one z-score) taller than the mean height of 175.0 cm, i.e. 181.5 cm. For the birth cohort 1959, the corresponding values are 6.5 cm (one z-score) taller than mean height of 179.1 cm, i.e. 185.6 cm.

We have provided these calculations in the Results section: “Supplementary file 2 shows that the mean heights range from 174.5 to 179.1 cm and the z-scores range from 6.4 to 6.6 cm for the birth cohorts in the total population without any clear patterns of change over time. The finding of 10% lower HR with one z-score higher body height suggests that men born in 1939 with a body height of 181.5 cm (equal to one z-score taller than the mean) can expect a 10% lower dementia risk compared with men with the mean body height (175.0 cm). The corresponding body height for the latest birth cohort (1959) is 185.6 cm compared to the mean body height of 179.1 cm.”

To make these calculations transparent, we have included the mean body heights in addition to the z-scores for each birth cohort in Supplementary file 2 for the full population, brothers and twins.

The authors discuss the limitations of obtaining dementia diagnoses from patient registers. However, they do not address the introduction of the national prescription register in 1995 and how that might affect sensitivity, specificity, and accuracy in identifying dementia cases.

Thank you for making us aware of this. We have now explored the relationship between body height and dementia before and after the introduction of the National Prescription Register that enabled identification of dementia diagnoses with ATC codes from January 1^st^ 1995. This is described in the following paragraphs in the manuscript:

In the Materials and methods section: “Sensitivity analysis of the association 1) before 1995 (dementia diagnosis identified by ICD-8 codes) and 2) 1995 and thereafter (dementia diagnosis identified by ATC codes and ICD-10 codes) was performed.”

In the Results section: “Analyses of Model 3 with the follow-up period divided in 1) prior to 1995 and 2) 1995 and thereafter showed that the association between one z-score taller body height and dementia was comparable between the two periods (Supplementary file 4).”

In the Discussion: “Furthermore, the introduction of the Danish Prescription Registry on January 1^st^ 1995 has enabled identification of dementia patients who are not in contact with the secondary healthcare system for their dementia diagnosis, but who receive medication through their general practitioner. This has most likely improved the sensitivity of the dementia diagnoses from 1995. Our sensitivity analyses showed comparable findings of the relationship between body height and dementia before and after the introduction, yet the comparison of these results were limited by including men in different age-groups (<1995: maximum age of 36-56 and ≥1995: maximum age of 57-77).”

Finally, we want to acknowledge that acetylcholinesterase inhibitors (ATC: N06D) may be used for patients with Parkinson’s disease, but the treatment is only indicated if the patients have comorbid dementia. Thus, this should not influence the specificity of the diagnosis. This information has been included as a foot note in Figure 1—figure supplement 1.

The authors include birth cohort in the analyses and compare results dividing at age 60. They do not, however, address the fact that participants were aged 57 to 77 at the latest date for register dementia diagnoses, so that a large proportion are younger than average age of onset for dementia. Prevalence of dementia is quite low, which likely reflects both cases missed by registers and the age structure of the study population.

We agree and have included the following statement in the strength and limitation section of the manuscript: “Also, the findings may not be generalizable to men aged above 77 years as the study population only reached a maximum age of 57-77 years during the follow-up period.”

We also agree that the low frequency of dementia in our cohort could be due to the particular age structure of the cohort, which implies that the data cannot be used to estimate prevalence, only incidence of dementia through this age range.

Not until the detailed methods is it learned that brothers were from 1953-1959 birth cohorts, making them age 57 – 63 as of the latest date of matching to dementia diagnoses. (The text states that the brothers were "mainly" from 1953-1959, so it's a bit ambiguous.) If the brothers are from 1953-1959 birth cohorts, it is not surprising that rate of dementia cases is lower for brothers than for the full population or for the twins. It seems possible that the difference in age structure might also contribute to the observation that the point estimate for the association between body height and dementia is greater in the twin pair analyses than in the analyses of brothers. It would also make the observation less interesting. This difference in birth cohorts is essential to note because, otherwise, the assumption would be that the sample of brothers contains all twin pairs in the sample of twins, since twins are also siblings.

We apologize for not being specific about the birth cohorts from which the brothers were included in the prior version of the manuscript. We have rewritten and provided more information of the birth cohorts where brothers were identified in the materials and methods section: “Brothers were identified from the population through maternal linkage in the Danish Civil Registration System and 73.5% were included from the birth-cohorts 1953-1959, where the linkage between mothers and offspring is complete in the register. The coverage vary between birth cohorts because the linkage between parents and offspring was based on shared residency in 1968 where the Danish Civil Registration System was established. After 1968, the linkage was carried out at the time of birth. Please see Supplementary file 1 for number of individuals included from each birth cohort (1939-1959).”

We want to highlight that even though 73.5% of brothers were included from the birth cohorts 1953-1959, 26.5% were included from the prior birth cohorts (1939-1952). As described in the paragraph above, this information is now available in the manuscript and Supplementary file 1.

Also, we want to clarify that the results from the within-brother and within-twin analyses, respectively, are compared to cohort analyses of the population of brothers and twins, respectively. Thus, the interpretation of these results is based on comparisons of results based on the same population: brothers and twins, respectively, and thereby identical age structures. It is thereby not possible that difference in age distribution can explain the difference identified in the within-brother/twin analyses from the cohort analyses of brothers/twins. We have clarified this in the Results section in the manuscript: “The analyses of Model 3 showed that the within-pair analysis of brothers revealed a stronger, but less precise, point estimate than the cohort analysis of brothers (Table 3). For twins, the comparison of the cohort analysis of twins and the within-pair analysis of twins revealed a pattern similar to that found for brothers. Yet, the pattern was less clear and the HRs were imprecisely estimated possible due to a lower number of individuals in these analyses (N=7,388) (Table 3)”

It would be of interest to describe the mean and distribution of intrapair difference in height in centimeters (cm).

The numeric mean and SE difference in body height between brothers and twins, respectively, as well as the distribution of these estimates are now provided in the Results section of the manuscript: “The numeric mean (SE) body height difference was 5.1 (0.02) cm between brothers and 3.5 (0.06) cm between twins (See Figure 2—figure supplement 1 for distribution of the numeric difference in body height for brothers (A) and twins (B)).”

Please note that brothers could include more than just two. In cases where there were more than two brothers with a shared mother, the difference in body height between all brothers were included e.g. for a groups of three brothers, the differences between brother one and two, brother one and three, and brother two and three, respectively, were included. This information is provided as a foot note in Figure 2—figure supplement 1.